# Characterisation of the Solidification of a Molten Steel Surface Using Infrared Thermography

**Carl Slater [1],\* , Kateryna Hechu [2], Claire Davis [1] and Seetharaman Sridhar [3]**

[1]  WMG, University of Warwick, CV7 4AL Coventry, UK; claire.davis@warwick.ac.uk
[2]  Tata Steel, Research & Development, 1970 CA IJmuiden, The Netherlands;
     Kateryna.Hechu@tatasteeleurope.com
[3]  George S. Ansell Department of Metallurgical and Materials Engineering, Colorado School of Mines, Golden,
     CO 80401, USA; sseetharaman@mines.edu
\*   Correspondence: c.d.slater@warwick.ac.uk; Tel.: +442476151577

**Abstract:** Infrared thermography provides an option for characterising surface reactions and their effects on the solidification of steel under different gas atmospheres. In this work, infrared thermography has been used during solidification of Twin Induced Plasticity (TWIP) steel in argon, carbon dioxide and nitrogen atmospheres using a confocal scanning laser microscope (CSLM). It was found that surface reactions resulted in a solid oxide film (in carbon dioxide) and decarburisation, along with surface graphite formation (in nitrogen). In both cases the emissivity and, hence, the cooling rate of the steel was affected in distinct ways. Differences in nucleation conditions (free surface in argon compared to surface oxide/graphite in carbon dioxide/nitrogen) as well as chemical composition changes (decarburisation) affected the liquidus and solidus temperatures, which were detected by thermal imaging from the thermal profile measured.

**Keywords:** liquid steel; non-contact measurement; oxides; steel-making

---

## 1. Introduction

With the ever-increasing demands on the steel industry to lower emissions and increase efficiency comes the equally demanding requirements for a higher quality product. This is leading to an increase in the number of sensors and monitoring devices used in the steelmaking process [1–3], whereby particular focus has been during the rolling process, where roll force, strip thickness, and even phase transformation after rolling (austenite to ferrite) can be monitored [4–8]. However, the conditions for higher temperature process monitoring are more challenging, for example, during casting or thermo-mechanical processing, but there is demand for real-time feedback to allow consistent and high quality steel to be produced. Some examples of where additional feedback related to steel surfaces would be useful for dynamic control include: the evolution of the solid shell during casting (morphology, phase, etc.), heat transfer during solidification and the rate/type of surface oxides forming in the ladle, which can be detrimental if entrained into the melt [9] or beneficial for lubrication during strip casting [10]. In addition, any ability to monitor scale formation during hot rolling of strip product would be beneficial, as the type, distribution and adhesion of the oxide scale influences the quality of the final product [11,12]. Identifying the type and distribution of molten and solid layers of slag on the steel surface in the ladle would be useful in controlling the steelmaking process, for example, if the slag location is detected correctly, the slag skimming process can be conducted with minimal steel losses, before pouring the molten metal into the tundish prior to casting [13]. Identifying if precipitation of solid phases occurs in the slag layer is also pertinent for controlling heat

transfer during continuous casting [14] and for potential recovery of valuable metals from steelmaking slags [15].

This paper explores the potential for using infrared thermography for surface characterisation during solidification under different gas atmospheres, to detect and differentiate changes resulting from reactions (formation of surface films) and/or phase changes (solidification) due to the differences in emissivity and latent heat. In order to induce physical changes, resulting from phase transitions and chemical reactions, on the liquid steel surface and during solidification, controlled changes in temperatures and/or gas atmosphere (argon (Ar), nitrogen ($N_2$) or carbon dioxide ($CO_2$)) were used.

## 2. Experimental

High temperature confocal scanning laser microscopy (CSLM) has been increasingly utilised for metallographic investigation since it was pioneered by Yin et al. [16]. In more recent years its usage for steel processing, particularly in relation to oxides and slags, has gained momentum [17,18]. The work reported in this paper applied the technique, developed by Slater et al. [19], of using combined infrared thermography with CSLM. The CSLM used was a Yonekura VL2000DX-SVF17SP (Yonekura, Yokohama, Japan) with a He-Ne laser as the imaging source. Due to the wavelength of the halogen heat source in the CSLM, a TM160 (Micro Epsilon, Birkenhead, UK) 7–14 μm infrared thermographer was used to reduce the amount of background noise.

Prior to testing, the CSLM was calibrated to ensure the melting temperatures recorded were accurate using standard size (2 mm × 2 mm × 2 mm) samples from four pure metals (Al, Au, Ni and Fe). Before the test, the CSLM was evacuated and backfilled with N6 argon three times, to ensure an initial inert atmosphere in order to avoid any surface reactions. A typical thermography image of a hot sample in the heated crucible can be seen in Figure 1. Here, due to the curvature of the droplet surface, only the very top could be seen clearly without scaling. An R type thermocouple was situated on the underside of the crucible; this temperature reading was used to provide calibration for the thermographer. During testing, radiant heat values were determined from the crucible and from the sample. Radiant heat was used due to the uncertainty and variability in emissivity values. By using an emissivity value, and its corresponding apparent temperature, then this reduces the uncertainty in emissivity, because should an emissivity be used that is slightly off, then the apparent temperature would change, still providing the same radiant heat. The radiant heat (Q) can be calculated from the following equation:

$$Q = A\varepsilon\sigma T^4 \tag{1}$$

where $\sigma$ is the Stefan–Boltzmann constant, $T$ is the apparent absolute temperature given by the thermographer, $A$ is the pixel area of the image and $\varepsilon$ is the emissivity.

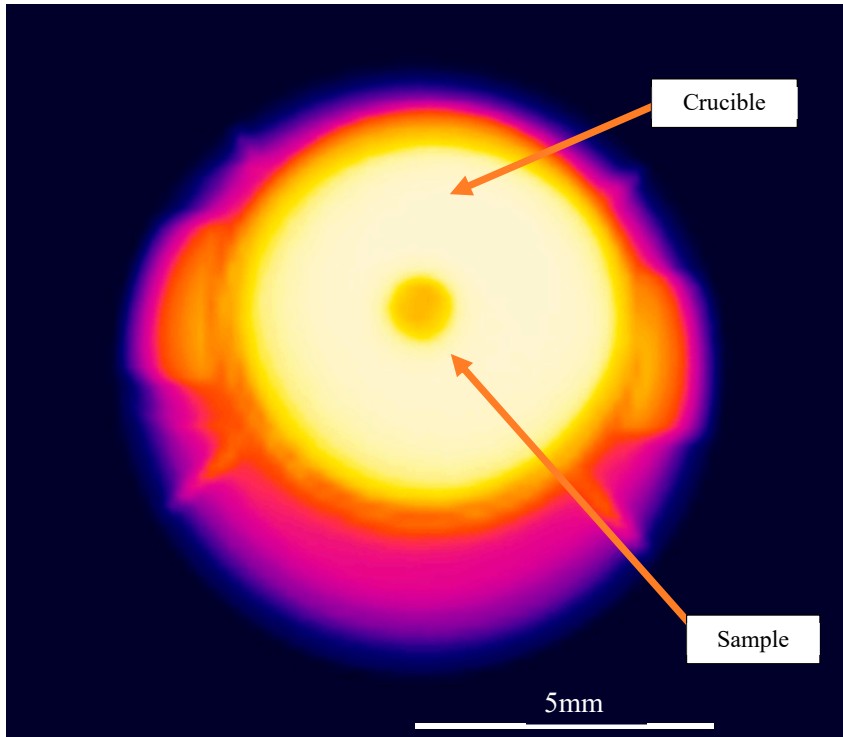

**Figure 1.** Typical thermography image taken from the confocal scanning laser microscope.

Three trials were carried out, all consisting of the same thermal schedule. A sample (2 mm × 2 mm × 2 mm) of TWIP steel (15% Mn 1.5% Al, 3% Si, 0.7% C) was placed inside an alumina crucible. The physical properties of the steel and crucible are given in Table 1, obtained using ThermoCalc (version 2018b, Stockholm, Sweden). The specimen was heated to 200 °C for 2 min, to dry the samples, before heating to 1500 °C at 10 °C/s and holding for 1 min. The heating was then turned off to allow natural cooling of the sample. All the trials were conducted using argon during the initial heating stage. However, during the dwell at 1500 °C, the atmosphere was either maintained as argon (Ar) or switched to nitrogen ($N_2$) or carbon dioxide ($CO_2$) at a rate of 200 mL/min.

**Table 1.** Summary of expected properties of the materials used in this study.

| Liquidus (°C) | Solidus (°C) | Emissivity of Solid Steel | Emissivity of Liquid Steel | Emissivity of Alumina |
| --- | --- | --- | --- | --- |
| 1380 | 1305 | 0.6 | 0.3 | 0.6–0.7 |

Once the CSLM reached 1500 °C, the crucible emissivity on the infrared thermographer was calibrated to the CSLM thermocouple (the crucible calibration was taken 500 μm away from the steel droplet). During dwell periods the temperature in the CSLM was much more homogenous and; therefore, best suited for setting the initial conditions. The infrared thermographer reading during cooling for all tests can be seen in Figure 2. Small differences near 1000 °C were due to small variations in the mass of the sample, as well as the difference in cooling capacity of the different gases. However, during the range of solidification (1380–1200 °C) very good repeatability can be seen. As the samples' surfaces changed emissivity, both during solidification and in the presence of a surface product, the stated temperatures in this paper refer to this calibrated point, and was, hence, why the distance away from the sample was limited to 500 μm. This method has shown very good agreement previously, where the impact of any thermal lag was shown to be negligible using differential calorimetry [19,20]. Therefore, the readings taken from the surface act to provide information on when events occur, rather than providing a direct temperature reading itself.

In all cases readings were taken from a single pixel of size 10 μm × 10 μm. The sample reading was taken from the apex of the liquid droplet.

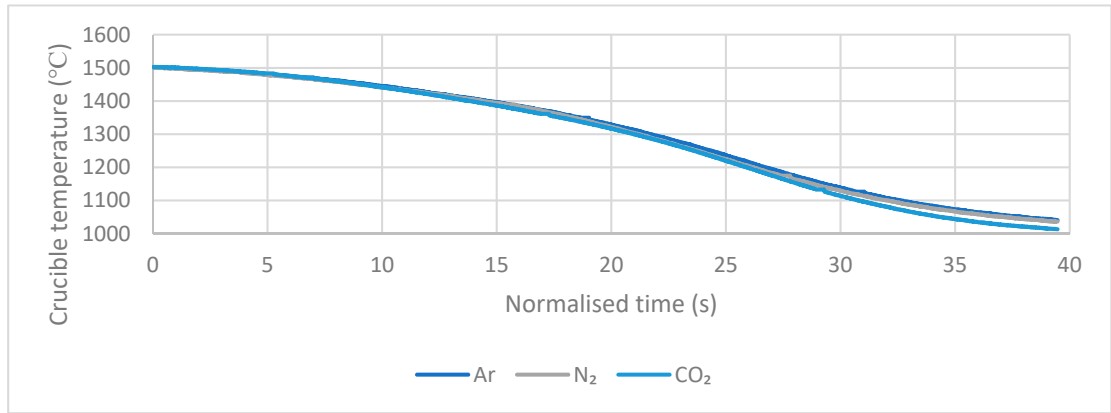

**Figure 2.** Temperature curves of the crucible during each test to ensure consistent conditions for each test. Normalised time refers to 0 s, being when the CLSM power was switched off.

## 3. Results and Discussion

In all three cases the radiant heat of the sample during the dwell at 1500 °C was 0.0033 W (±0.0002), an emissivity of around 0.08 was measured during the dwell. This was lower than seen in literature; however, the quality of vacuum in the CSLM was very high and likely to have less oxidation than a conventional exposed liquid steel surface.

The first condition observed was that of a continual argon atmosphere, which can be seen in Figure 3. Once the power of the heating was turned off (at about 5 s), a gradual decrease in the radiated heat could be seen due to the cooling of the sample. After around 20 s, the sample emittance dropped quickly (convection ripples on the liquid surface due to the formation of solid subsurface) before an increase could be seen, which was attributed to both the latent heat of fusion as well as the higher emissivity that the solid surface exhibited in comparison to the melt. After solidification, the slope of the radiated heat was steeper than prior to solidification, suggesting the higher emissivity was resulting in a greater cooling rate of the sample. As the thermal mass of the sample was much greater than that of the crucible (600 mg compared to 200 mg), then the cooling of the system was much more heavily dictated by the steel. Figure 4 shows CSLM images taken from the sample at specific time intervals from Figure 3, confirming that the sample was partially solidified at 23 s and full solidification had occurred at 30 s.

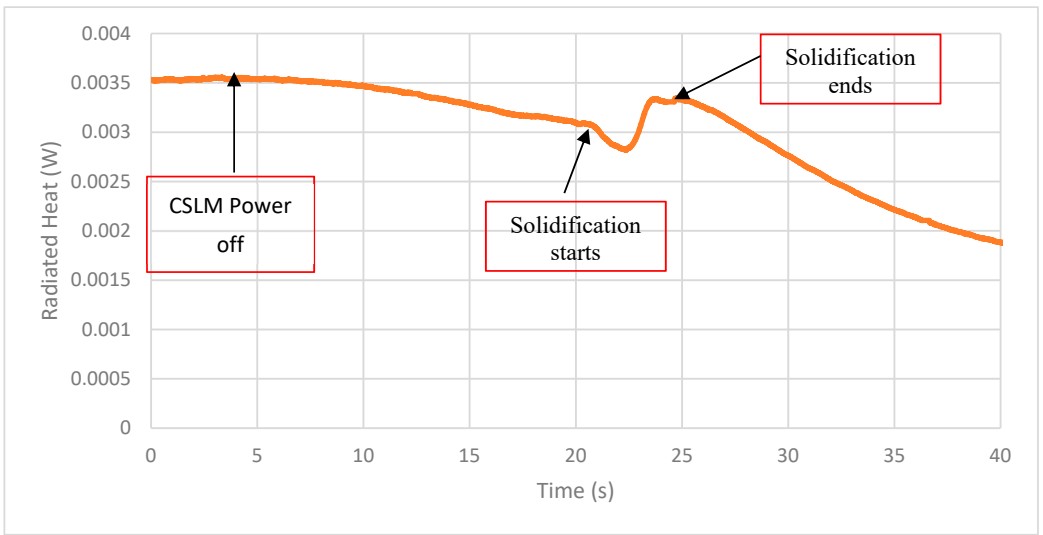

**Figure 3.** Radiated heat curve for the surface of the steel sample that was solidified in Ar.

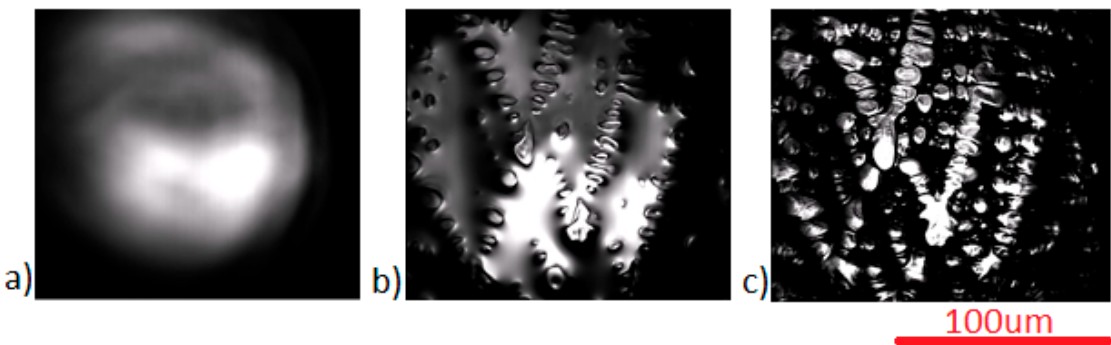

**Figure 4.** Images from the CSLM during cooling in Ar, taken at (**a**) 5 s (liquid steel), (**b**) 23 s (partial solidification) and (**c**) 30 s (fully solidified) according to Figure 3.

The second trial used a $CO_2$ atmosphere during the cooling of the steel (Figure 5). For this sample an initial fluctuation in the radiant heat could be seen in the sample as soon as the gas was changed (at approximately 5 s), this was followed by a gradual increase in radiated heat due to the formation of an oxide layer that had a higher emissivity. Turning the heating power off was followed by a gradual decrease in radiated heat as the sample cooled. A change in gradient was observed at approximately 33 s, when solidification started, but the effect of solidification on the curve was much smaller than observed during solidification under the argon atmosphere, Figure 3. This reduced signal from solidification was to be expected, as the increase in radiated heat was now only attributed to latent heat and not to a change in the emissivity of the surface, as there was already a solid oxide film present. Figure 6 shows the CSLM images taken from the $CO_2$ sample. Prior to the $CO_2$ introduction, the sample showed the reflective surface of the liquid steel droplet; however, once the $CO_2$ gas was introduced, then oxide formed, as can be seen in Figure 6b. It was seen from the continuous CLSM imaging that a complete oxide film formed over the surface within 0.5 s from the $CO_2$ gas being introduced. Therefore, the gradual increase in radiant heat, expected from an increase in emissivity, from the moment the $CO_2$ gas was switched on suggests that the thickness of the oxide was increasing with time.

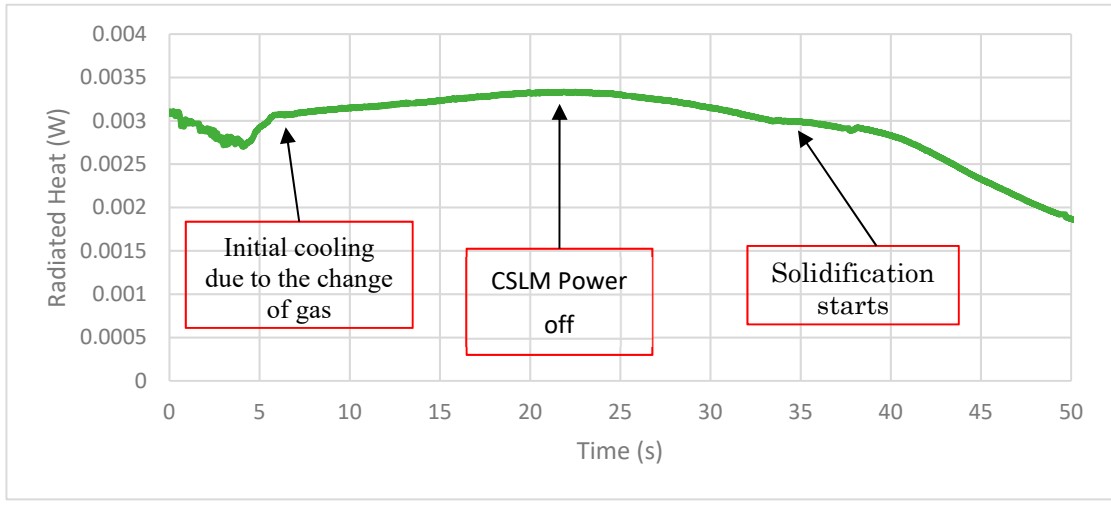

**Figure 5.** Radiated heat with time of the steel surface during cooling in a $CO_2$ atmosphere.

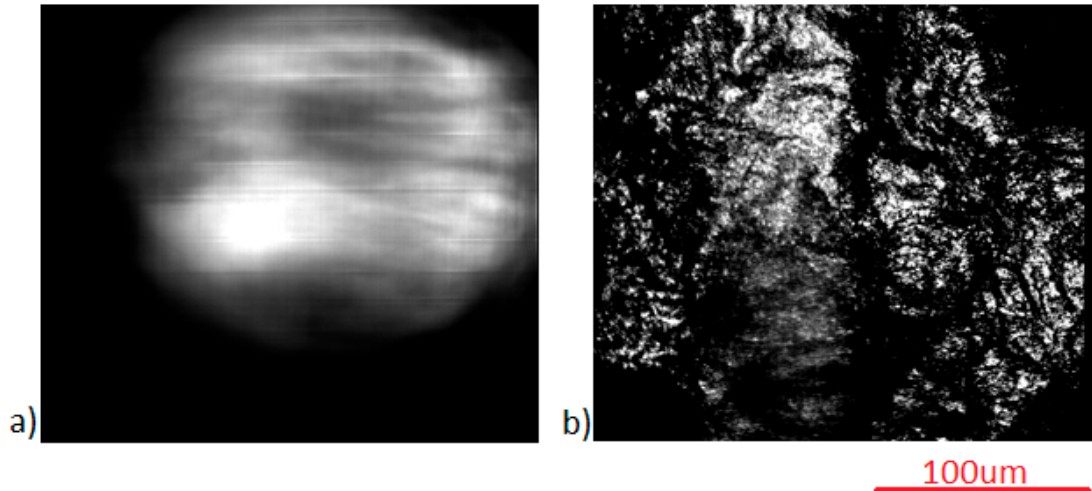

**Figure 6.** CSLM images taken from the sample cooled in a $CO_2$ atmosphere, taken at (**a**) 0 s (liquid steel) and (**b**) 25 s (oxide film) according to Figure 5.

The final sample was tested under a nitrogen atmosphere. Nitrogen has previously been seen to cause and stabilise the formation of graphite on the surface during decarburisation of the bulk steel [21]. It was suggested by Slater et al. [21] that this mechanism is a result of nitrogen reducing the solubility of carbon in the liquid metal, thus stabilizing graphite. Once graphite has formed, nitrogen can then further react with the graphite to produce a combination of C2N2 (cyanogen) and XCN (variable cyanides). This more dynamic reaction with the surface, compared to argon, is likely to impact the heat flux of the surface.

Figure 7 shows the radiant heat curve for this sample. It can be seen that there was a significant increase in emitted heat during the initial stage of gas interaction compared to the other samples, with the radiant heat increasing by more than $10\times$ that of the sample held in $CO_2$ during the same period. This was due to the formation of graphite on the surface, shown in Figure 8, which has a higher emissivity than the liquid steel and of an oxide, approaching that of a black body [1]. Incomplete coverage of the surface was seen, where the amount of graphite can be seen to increase from 0 to 30 s. Due to the scale of the graphite, the radiated heat in this case refers to the average within a pixel (approximately 100 $\mu m^2$); therefore, as surface coverage increased, the overall emissivity of a pixel increased also, thus giving feedback into the rate of reaction on the surface. At 30 s the reaction rate had reduced considerably (little change in graphite coverage of the sample), this may be due to importance of the Fe–N interaction in reducing the solubility of carbon, and thus could not reach a point where the whole surface was coated in graphite. Once the power of the CSLM was switched off, a large decrease in radiant heat was observed as the sample cooled quickly, and solidification could also be seen (at approximately 45 s), although the change in gradient was not as pronounced as for the sample in an argon atmosphere.

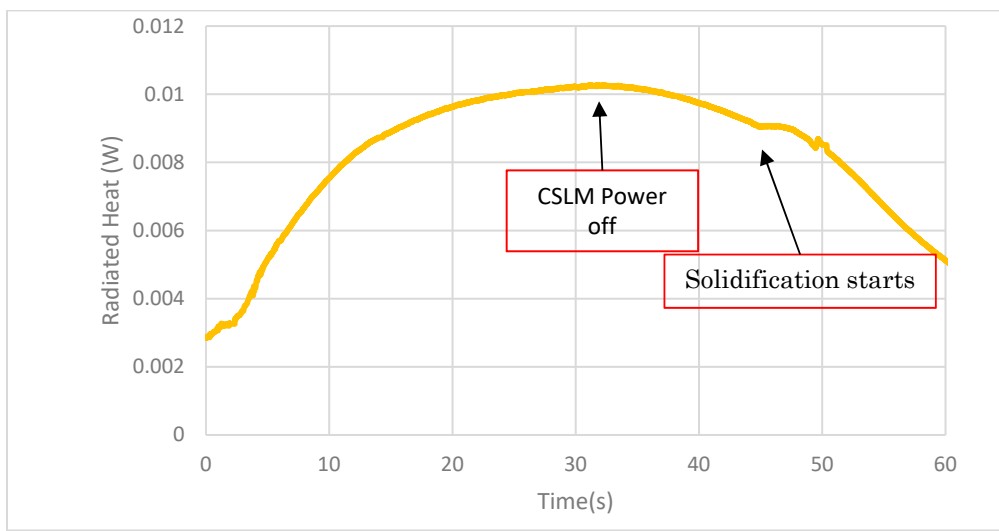

**Figure 7.** Radiant heat with time taken from the surface of a sample solidified in a $N_2$ atmosphere.

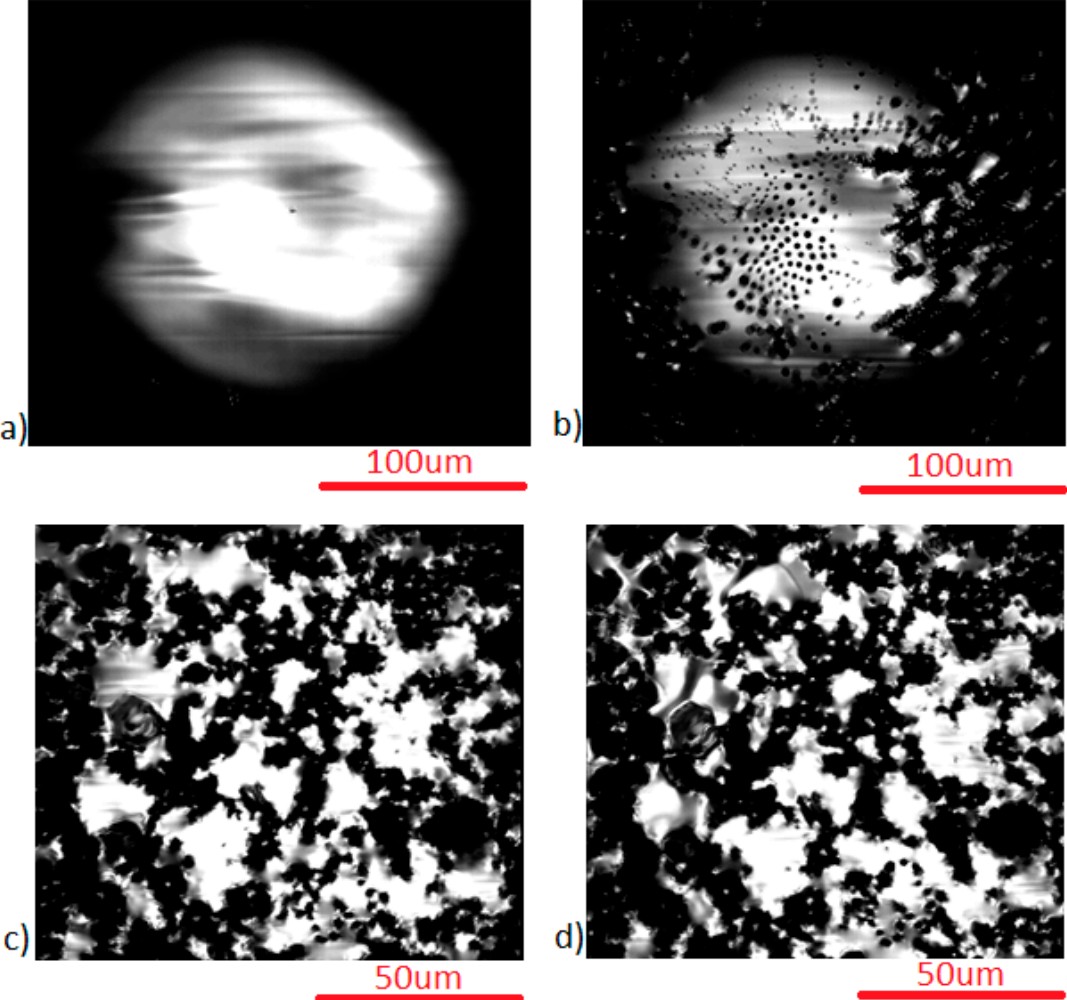

**Figure 8.** CSLM images taken during the solidification of steel under a $N_2$ atmosphere at (**a**) 0 s (liquid steel), (**b**) 10 s (graphite flake on steel surface), (**c**) 30 s (increase in graphite flakes) and (**d**) 45 s (start of solidification–dendrite interfaces arrowed).

Table 2 shows a summary of the solidus and liquidus temperatures of the steel solidified, under natural cooling, in the CLSM, in the three different atmospheres. It can be seen that the cooling under an argon atmosphere gave a liquidus much lower than that of predicted equilibrium solidification, which was expect as nucleation was limited from the free surface (the high purity of the atmosphere means that no inclusions or surface oxides were observed) and; therefore, a large undercooling was required locally before solidification was seen. This is consistent with previous work on the CSLM, where the smooth crucible surface and small sample size can reduce the chance of a nucleation event significantly [20]. Under a $CO_2$ atmosphere, the newly formed surface oxide can act as nucleation sites, and this minimises the undercooling required to a point where the liquidus matches very closely to the equilibrium predicted value (it is also possible some carbon was removed during oxidation, which would increase the liquidus; this is expected to be a minor influence as oxygen is predicted to more favourably bond with Al and Si, compared to forming CO (Factsage 7.2, Aachen, Germany)). Finally, the nitrogen atmosphere significantly increased both the liquidus and solidus temperatures, which can only be achieved through a dynamic change in the composition. As graphite forms on the surface, this decarburises the steel and thus increases the liquidus and solidus temperatures. If 0.5% carbon was removed from the bulk then the liquidus and solidus would have been increased to 1412 °C and 1355 °C, respectively. This level of decarburisation has been shown previously through a nitrogen atmosphere in a CSLM [21].

**Table 2.** Summary of the solidus and liquidus temperatures of the steel, determined from thermography in the three different atmospheres, and the predicted (from Thermo-Calc) values.

| Equilibrium | Liquidus (°C) | Solidus (°C) |
|---|---|---|
| | 1380 | 1305 |
| Ar | 1323 | 1279 |
| $CO_2$ | 1373 | 1283 |
| $N_2$ | 1402 | 1366 |

## 4. Conclusions

The work presented has highlighted that infrared thermography can be used to obtain information about surface reactions and solidification temperatures even when the liquid steel cannot be directly imaged. Specifically, TWIP steel samples were solidified under different gas (Ar, $CO_2$, $N_2$) atmospheres in a confocal laser scanning microscope, and infrared thermography was used to monitor the sample surface.

Different surface conditions were observed (simple liquid steel in Ar; oxidation in $CO_2$; decarburisation and graphite flake formation in $N_2$), which affected the emissivity and, hence, the radiated heat detected by thermography. The different surface reactions could be characterised by their different radiative properties.

Solidification of the steel could be detected from changes in the radiated heat signatures of the sample, even when a surface film (oxide or graphite) was present. Differences in the liquidus and solidus temperatures were related to changes in local chemistry (decarburisation in $N_2$) and nucleation (oxide and graphite) conditions.

The work has indicated that infrared thermography may be a suitable non-contact measurement method to provide information in areas such as the tundish in steel making (due to any emissivity changes of the slag material as it reacts with the steel) or the hot strip surface, as steel comes from the mould in a continuous caster in defining blowouts etc.

**Author Contributions:** C.S. and S.S. conceived and designed the experiments; C.S. performed the experiments; C.S., K.H., C.D. and S.S. analyzed the data; C.S. wrote the paper.

**Funding:** This research was funded by EPSRC, grant number EP/M014002/1.

**Acknowledgments:** The authors would like to thank EPSRC for funding and also WMG for their support and facilities.

**Conflicts of Interest:** The authors declare no conflict of interest.

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
