# Peer review of "Characterisation of the Solidification of a Molten Steel Surface Using Infrared Thermography"

_metals, doi:10.3390/met9020126_

Round 1
Reviewer 1 Report
This work presents a very interesting and probably promissing technique to investigate surface phenomena at high and very high temperatures where other methods can not be used. However, many important data are lacking in the manuscript that makes the explanations proposed by the authors very questionable. Below is the main points which need to be properly discussed and clarified before publication.
1. The title of the paper is too general and vague. What was the main goal of characterization: surface reactions ? solidification ? radiant heat flux ?
2. Chemical composition of TWIP steel is indicated in a vague matter. Please specify the composition, especially for Al and C concentration.
3. The authors used ThermoCalc and Comsol Multiphysics to obtain physical properties of steel. These two softwares are used for quite different purposes. What properties were obtained with the use of Comsol Multiphysics ?
4. Emissivity, especially of liquid steel, is dependent on the steel chemical composition. The same can be said about oxide film formed on the steel surface. How did the authors manage this issue ? Are the emissivity values in Table 1 related to the TWIP steel ?
5. The effect of nitrogen is very interesting but two points at least should be properly explained. The first one is the carburization mechanism. Are there any data (chemical analysis, etc) which confirm that the black spots are graphite ? The second point is as follows. The emissivity of graphite can not exceed 1. Therefore, it is difficult to explain the 10 fold increase in radiant heat flux compared to the CO2 atmosphere.
It is my opinion that such reactions as chemical adsorption of CO2 and N2 on the melt surface can affect the net radiant heat flux from the surface. At least this mechanism should be also discussed.
Author Response
Thank you very much for your response. I very much appreciate that you see some interesting benefits behind this technique. I agree with your comments and have amended as such.
1) The title is now"Characterisation of the solidification of molten steel surface using infrared thermography."
2)I have been more specific with the composition. I was awaiting clearance for the supplier for me to allow to publish the exact values
3)Sorry that was my mistake. I did some work on Comsol on heat flow and cooling rates but did not end up putting it in. I have omitted the mention of COMSOL.
4)This is the reason for converting back to radiant heat. The thermal camera gives a temperature for a given emissivity. even if the emissivity was wildly out, then the measured temperature would change accordingly and give the same radiant heat value. This does not let me split temperature and emissivity changes as mentioned in the paper but does reduce the error. With the thermocouple in the CSLM I am also able to get a fairly close reading for emissivity anyway.
5) You are correct and thank you for pointing this out. The actual emissivity of the liquid in these cases was around 0.08. i had not mentioned that and i have now added this. The reason it is lower than literature liquid steel emissivities is due to the very high purity Ar atmosphere we have in the CSLM compared to the standard use of IR measurements of steel which is usually in the ladle or during a pour. it is hard at this point to explore much of the N2 reaction but it is our purpose to do so. At the moment by the time the steel has cooled, either the C has diffused back into the solid or has cracked and fallen off the surface, and thus hard to gain much more insight. I have cited a reference to a previous paper where I have tried to give the suggested method of the reaction.
Thank you again for the time you have spent reading this.
Reviewer 2 Report
The paper describes in situ observation of the solidification of a TWIP steel in a CSLM under different atmospheres. The observation is accompagnied by thermographic measurements. The influence of the atmosphere on the solidification processdepends on reactions of the atmosphere with the steel and reflects in the thermal behaviour as well as in the surface structure.
The paper is written well, but should be improved in some points listed below.
Line 8: “using a Confocal Scanning Laser Microscopy”
replace by: “using Confocal Scanning Laser Microscopy”
or “using a Confocal Scanning Laser Microscope”
Line 20 [1]–[3] replace by [1-3]
Line 22: [4]–[8] replace by [4-8]
Line 31: [11], [12] replace by [11, 12]
Line 46 (CLSM) replace by (CSLM)
Line 47: Emi et al replace by Yin et al (Citing the first author)
Line 48/49: [17], [18] replace by [17, 18]
Line 71: The CSLM was heated to 200 °C - Here it would be better to write:
The specimen was heated to 200 °C
Line 72: The CSLM power was then turned off better: The heating was then turned off
Line 75: 200 mL/min. replace by 200 ml/min.
Line 89: [19], [20] replace by [19, 20]
Line 90: delete °
Line 98 Once the power has been turned off in the CSLM better: Once the power of the heating has been turned off
Line 101: Is the higher cooling rate after solidifaction only affected by the higher emission or is there also an influence of a greater surface due to the structure on the surface formed during the solidification process (see figure 4)
Line 118: The CSLM power off better: The heating power off
Figure 5: “Initial cooling due to change of gas” The arrow from
this text box points to a step with increasing Radiated heat. That is
not understandable.
Is the temperature kept constant between 5 and
22 s? Otherwise, the change of Radiated heat can be contributed to a
heating of the specimen.
Line 149: Once the power of the CSLM is switched off better: Once the power of the heating is switched off
Figure 7: Is the temperature kept constant between 0 and 31 s? Otherwise, the change of Radiated heat can be contributed to a heating of the specimen.
Line 165: pervious replace by previous
Line 171: What means „Factsage 7.2“?
Line 229: Al2O3/SiO2 replace by Al2O3/SiO2
Line 232 FeO-CaO-SiO2 Slag replace by FeO-CaO-SiO2 Slag
Line 234: "In-situ" replace by In-situ
Line 237 Al2O3-CaO replace by Al2O3-CaO
References: It would be good to add the DOI
Author Response
Thank you very much for your feedback. It was all straightforward and i have corrected everything you have mentioned.
The only things i think i need to reply with are:
Factsage is a thermodynamic package that helps define oxide stability.
You mention that does the temperature change during graph 7. No all the graphs were normalized to once the hold temperature was reached. I have made this clearer now.
Thank you for your time in checking this.
Reviewer 3 Report
The introduction does not sufficiently point out desired connection of the actual work with real steel casting. It is a rather general section about somehow desired process conditions and wanted process information. In addition, in which steel casting process does one have optical access to the solidification front of the steel to investigate it with a thermography system and use it as an input for some control loops? Introduction and conclusions target on the characterization or detection of the solidification process. Normally, the solidifying steel shell is hidden by some kind of mould and the liquid steel is coverd by a thick slag layer.
What is the purpose or motivation of using the different gas atmospheres? Which steel casting process is really supplying the free surface with one of the three gases? Please state more clearly the purpose of the gases in the experimental section, e.g. to get the different surface reactions and surface layers. This information should be given already in the experimental section.
Figure 1: The length scale in the Figure bottom is stating 5 metres and its text is partially covered by the white reference length! I guess this isn’t the actual size when the sample has only dimensions of 2 mm. Please indicate the temperature reference or calibration point, which is stated in the text, also in this picture. What is the scale / physical meaning of the colours? A corresponding colour-bar is missing.
The measurement point or measurement area for the values from the steel sample is not exactly given. Are the curves in Figure 3, 5 and 7 the product of some spatial averaging, are the values taken from a single point in the centre of the sample or anything else? If the results were taken from an area, how big is this area?
Please describe in more detail in the experimental section, how you determine the different phases of solidification in the sample over time. Although one might have a guess, it is not explained anywhere. At the moment, the arguing and the sequences in the paper sounds a little bit like it is obvious just from the curves of the radiated heat itself and the microscopy images are just a proof for that.
A suggestion: The temperature curve of the steel sample might be also helpful in Figure 3, 5 and 7. This should make it is possible to discover the values of Table 2 already in these plots and to see more clearly the solidification stages of the sample.
The transfer from a temperature curve measured by the thermography system like in Figure 2 to a curve of radiated heat like in Figure 3, 5 and 7 is not becoming entirely clear. According to the given equation, the emissivity has to taken into account. However, this requires already the knowledge of the underlying state of the material, doesn’t it? But according to the text, the material was deduced from the graph of the radiated heat. So it seems to be somehow a circular logic.
The Table 1 is giving the emissivity of liquid and solid steel. The text states changes in radiated heat due to different emissivity of other materials, like oxides (line 118) or graphite (line 141). However, the values for emissivity of these materials are missing in Table 1.
Please distinguish between the microscope and the heating mechanism. I suspect from working principle of the microscope, that not every CLSM is equipped with an additional heating source originally, isn’t it? It is always stated in the text and the figures, that the microscope (CLSM) is switched off. Surprisingly, there were nevertheless presented pictures from samples, after the microscope was switched off.
The conclusion in line 146 about a steady state condition is ambiguous. The authors relate the plateau to the Iron – Nitrogen interaction. However, this plateau coincides with the powering off of the CSLM.
Some statistics in table 2 – like a standard deviation – would suit very nicely. How many samples were investigated for each of the configurations?
Last sentence of the conclusions: The connection to continuous casting seems to be a little bit far fetched. The steel shell has already grown to several mm or even cm thickness when the strand leaves the mould. A relation of surface thermography on the underlying solidification processes deep inside the strand is for my understanding not supported by this work. Further, the steel shall remain liquid in the tundish and a complete slag layer on the free surface shall protect the steel from the atmosphere. The thickness of the slag layer in the tundish is much bigger than the thin reaction layer of the samples of this study. Therefore, the slag is a barrier prohibits the access to the liquid steel by the thermography and a transfer from results of this paper to a tundish are uncertain. Additionally, the missing solidification at the tundish is also a missing link to this study.
Additional minor remarks
The affiliation of the authors is missing.
Please consider rewriting the long sentences with several parentheses and slashes at the same time. This makes the reading and understanding of these sentences unnecessarily difficult. One example is already the abstract (line 11 ff.).
Please check the text layout again. There are multiple double-spacings in the text.
Please check grammar in the sentence of line 21. There are only auxiliary verbs (“… has been …”) and the main verb seems to be missing.
Author Response
Hello and thank you very much for your indepth review of this paper.
The work for this paper has come from work on horizontal belt casting where the top surface is exposed to the atmosphere and different gases are used. However, currently several patents are being processed and as such a paper on the technique of thermography as a way to measure aspects that is difficult by other methods has been proposed here. The field of belt casting is growing quickly and a paper on the gas interaction is of keen interest. However in a broader sense then thermography use in welding, soldering etc is also useful, but as these are more complex systems then the basic concept of monitoring of a metal surface solidifying is covered by this technique.
Apologies. The image text has been cropped in the reshuffle. the length scale is 5mm and has been corrected. The colour means nothing in this image as no calibration has been carried out. Calibration takes place empirically with the extracted data. The image acts to show the size of the system.
The measurement area is a single pixel of size 100x100um from the centre of the sample. I have added this to the text.
Due to the emissivity changes then actual temperature measurement is not possible. Emissivity is sensitive to phase, composition and any contamination on the surface. Radiant heat has been used as this removes any error in emissivity values, however, this means that exact temperatures cannot be determined. As such the temperature was read from the crucible temperature. I believe including this temperature on the graph may confuse people into thinking the thermography can give the temperature. As such, i only talk about real temperatures later including the error of using the crucible temperature. The key to these graphs is the shape, and the changes seen.
Figure 2 is easy to give a temperature as this is for alumina and has a set emissivity. I can also calibrate this against the TC in the CSLM. For the rest of the curves then equation 1 was used, where the radiant heat can be calculated based on the emissivity and temperature. Both used in the measurement. If the user defines the emissivity as one in the measurement then the temperature readout will not give a true value, but the pairing of the two values will give a true radiant heat. i.e using a value of 1 may give a temperature of 1250C whereas a value of 0.5 will give a temperature of 1450C. Both will give the same radiant heat and thus removes the need for specific emissivity values.
Actually, the CSLM is the heating platform in this case. All imaging has been through the thermographer. The image from the thermographer always looks the sample as figure 2 as the color is auto scaled, hence no further images. The method of CSLM is well known and covered in the literature also by the work by Yin et al.
Line 146, the rate of change of the curve is slowing prior to the CSLM being turned off showing the reaction slowing down. It is mentioned that "little" change can be seen and we are not stating that it is complete. I have now changed this to "the reaction rate has reduced considerably" to clarify this point.
i have been more specific in the conclusions that you have mentioned an now reads. The work has indicated that infrared thermography may be a suitable non-contact measurement method to provide information in areas such as the tundish in steelmaking (due to any emissivity changes of the slag material as it reacts with the steel), or the hot strip surface as steel comes from the mould in a continuous caster in defining blowouts etc."
Both of these we have got initial results for and show promising. However the surface of the strip as it comes out of the concaster is not too thick to pick up the changes, however due to the temperature profile within the strip then the curves are much flater and less pronounced, hence needing a base level understanding from this paper to understand the signal from a real casting scenario.
I would like to thank you again for the time you have spent reading through this. I very much appreciate your comments and the paper is certainly the better for it.
Round 2
Reviewer 1 Report
I requested to clarify the following 5 questions however, at least, three of them, 3,4 and 5 remain unanswered. Besides, the manuscript contains a lot of grammatical errors
1. The title of the paper is too general and vague. What was the main goal of characterization: surface reactions ? solidification ? radiant heat flux ?
2. Chemical composition of TWIP steel is indicated in a vague matter. Please specify the composition, especially for Al and C concentration.
3. The quastion was a follows
The authors used ThermoCalc and Comsol Multiphysics to obtain physical properties of steel. These two softwares are used for quite different purposes. What properties were obtained with the use of Comsol Multiphysics ?
In the new version of manuscript there is no mention about Comsol Multiphysics. Do the authors mean that all properties were calculated with ThermoCalc ? However, ThermoCalc does has any option to calculate emissivity. Please clarify.
4. Emissivity, especially of liquid steel, is dependent on the steel chemical composition. The same can be said about oxide film formed on the steel surface. How did the authors manage this issue ? Are the emissivity values in Table 1 related to the TWIP steel ?
5. The effect of nitrogen is very interesting but two points at least should be properly explained. The first one is the carburization mechanism. Are there any data (chemical analysis, etc) which confirm that the black spots are graphite ? The second point is as follows. The emissivity of graphite can not exceed 1. Therefore, it is difficult to explain the 10 fold increase in radiant heat flux compared to the CO2 atmosphere.
It is my opinion that such reactions as chemical adsorption of CO2 and N2 can affect the net radiant heat flux from the surface. At least this mechanism should be also discussed.
Typical grammatical errors
Line 68: Typical thermography image take from the CSLM
Line 89 … and hence why the distance away from the sample is minimised to 500 μm
Line 100~104 The sentence is too long and hard to understand
In all three cases the radiant heat of the sample during the dwell at 100 1500 °C was 0.0033 W, (±0.0002), an emissivity of around 0.08 was measured during the dwell, this is lower than seen in literature, however the quality of vacuum in the CSLM is very high and unlikely to be a similar condition to of oxidation to conventional exposed liquid steel surface.
Line 128 : figure 3 (Capital letter shoud be used)
Line 139 : Radiated heat profile… (Why “profile”? The graph shows a time variation)
Line 149 : emissivity than the liquid steel, and of oxide
Author Response
Hello,
Yes apologies i did not respond to some of your points:
1) I have removed comsol listed as i had thought to put in some work on the heat conduction through the sample. Sorry if it is still uncertain. Although emissivity is indeed an important parameter, however is is not a static value. as you said it changes with composition but also temperature and any turbulence on the surface. What thermal imaging cameras actually measure is emited IR radiation. It takes this signal and applies the emissivity to give temperature. What i have tried to do in this study is remove the need of emissivity as it is changing (although the affect it has on cooling rate can be seen). For a given value of emissivity then the thermographer gives an apparent temperature. If i chose a different emissivity, then i would see a different apparent temperature. However, both these would give the same value of emitted heat and this is what i have done here. To further clarify this i have changed the equation to clearly show that the temperature is the apparent temperature from the thermographer. I hope this clarifies things for you
2) see above
3) Yes you are correct and i have now added a paragraph on the mechanism. You are also correct in suggesting that this more dynamic reaction with the surface will influence the heat flux of the surface. This will not show its self until cooling, as prior to this the CSLM is under temperature control and the sample will remain a stable temperature whilst we record the surface emittance.
Thank you very much for spotting the grammatical errors.
Reviewer 3 Report
Thanks for the commments and explanations.
Regarding the thermography: The convertion of the thermography image (pixel) to the wanted radiated heat should be explained a little bit in more clearly in the paper. It might be worth considering to put some of your explanation from the "authors response" about the issue of emissivity and temparature into the paper, or to give a literature refenece, at least.
The pixel size in your comments (100 µm) and in the paper (10µm) differs by one order of magnitude. Please check for the correct value.
Author Response
You are correct. thank you. I have now added a paragraph in the experimental explaining the conversion and why radiant heat was used and not a constant value of emissivity..Thank you for your comments.
Sorry the pixel size is indeed 10x10um and thus an area of 100um2. My mistake in the comments back to you.